# Human antibodies targeting Zika virus NS1 provide protection against disease in a mouse model

Mark J. Bailey [1,2], James Duehr[1,2], Harrison Dulin[3], Felix Broecker[1], Julia A. Brown[1,2], Fortuna O. Arumemi[4], Maria C. Bermúdez González[1,5], Victor H. Leyva-Grado[1], Matthew J. Evans[1], Viviana Simon[1,5,6], Jean K. Lim[1], Florian Krammer [1], Rong Hai[3], Peter Palese[1,7] & Gene S. Tan[4,8]

Zika virus is a mosquito-borne flavivirus closely related to dengue virus that can cause severe disease in humans, including microcephaly in newborns and Guillain-Barré syndrome in adults. Specific treatments and vaccines for Zika virus are not currently available. Here, we isolate and characterize four monoclonal antibodies (mAbs) from an infected patient that target the non-structural protein NS1. We show that while these antibodies are non-neutralizing, NS1-specific mAbs can engage FcγR without inducing antibody dependent enhancement (ADE) of infection in vitro. Moreover, we demonstrate that mAb AA12 has protective efficacy against lethal challenges of African and Asian lineage strains of Zika virus in $Stat2^{-/-}$ mice. Protection is Fc-dependent, as a mutated antibody unable to activate known Fc effector functions or complement is not protective in vivo. This study highlights the importance of the ZIKV NS1 protein as a potential vaccine antigen.

[1] Department of Microbiology, Icahn School of Medicine at Mount Sinai, New York, NY 10029, USA. [2] Graduate School of Biomedical Sciences, Icahn School of Medicine at Mount Sinai, New York, NY 10029, USA. [3] Department of Plant Pathology and Microbiology, University of California Riverside, Riverside, CA 92521, USA. [4] Infectious Diseases, The J. Craig Venter Institute, La Jolla, CA 92037, USA. [5] The Global Health Emerging Pathogens Institute, Icahn School of Medicine at Mount Sinai, New York, NY 10029, USA. [6] Division of Infectious Diseases, Department of Medicine, Icahn School of Medicine at Mount Sinai, New York, NY 10029, USA. [7] Department of Medicine, Icahn School of Medicine at Mount Sinai, New York, NY 10029, USA. [8] Department of Medicine, University of California San Diego, La Jolla, CA 92037, USA. Correspondence and requests for materials should be addressed to G.S.T. (email: gtan@jcvi.org)

Zika virus (ZIKV) is an arthropod-borne flavivirus closely related to dengue, yellow fever and West Nile viruses, which has caused an emerging epidemic in the Americas, the Caribbean, and the Pacific regions. While ZIKV is spread primarily through the bite of an infected *Aedes* species mosquito, cases of sexual transmission have also been reported[1,2]. ZIKV infection is associated with severe illness in humans including microcephaly and birth defects in newborns[3–5] and Guillain-Barré syndrome in adults[6,7]. Consequently, ZIKV infection poses significant threats to global health.

To understand the molecular determinants of immunity to ZIKV infection, several groups have isolated monoclonal antibodies (mAbs) from patients infected with ZIKV[8–12]. These studies have revealed important antigenic sites on the envelope (E) protein required for virus neutralization. Quaternary epitopes such as the "envelope dimer epitope", which are dependent on the native dimeric assembly of the E protein, are promising vaccine and therapeutic targets, as mAbs generated against these sites tend to be potently neutralizing[10]. However, one chief concern in the development of flavivirus vaccines targeting the E protein is the phenomenon of antibody-dependent enhancement of disease (ADE). This occurs when viral replication is enhanced by pre-existing antibodies that opsonize but do not fully neutralize the virion resulting in enhanced uptake of the virion-antibody complex by FcγR-bearing target cells. The virus is then able to replicate in these cells, increasing the severity of disease[13]. Though there is no epidemiologic evidence that Zika virus can cause ADE in humans, studies have shown ZIKV-induced monoclonal antibodies targeting the E protein can enhance infection of ZIKV or DENV in vitro and induce mortality in DENV-infected mice[8]. Additionally, passive transfer of DENV or WNV immune plasma to immunocompromised mice has resulted in more severe disease progression upon ZIKV infection in vivo[14]. Consequently, ADE may limit the therapeutic application of E protein-specific antibodies and vaccines against Zika virus.

Other viral proteins including non-structural 1 (NS1) protein have emerged as promising targets as antibodies that do not bind the virion are unlikely to enhance disease. In a recent study of four patients infected by ZIKV, 34.4% of virus-specific mAbs target the NS1 protein[8]. This immunogenic glycoprotein plays an essential role in viral RNA replication and immune evasion. The NS1 protein is initially translated as a monomer, becomes glycosylated in the ER and subsequently forms a dimer that can potentially traffic to multiple distinct locations within the cell[15]. The NS1 protein of many flaviviruses is known to associate with the viral replication complex on the surface of the endoplasmic reticulum membrane, associate with the plasma membrane by a glycosylphosphatidylinositol linker, exit cells to form a lipophilic hexamer, and potentially bind to uninfected cells via glycosaminoglycan interactions[16].

Protective antibodies against viral pathogens are able to protect via multiple mechanisms: neutralization, Fcγ-receptor mediated viral clearance, and complement-dependent cytotoxicity (CDC)[17]. Antibodies against the NS1 protein were shown to be protective against a number of different flavivirus species. In Japanese encephalitis virus, NS1-specific antibodies were found to reduce viral output from infected cells[18]. Yellow fever virus NS1 fragments were used as a vaccine and immunized mice had reduced neurovirulence upon viral challenge[19]. Later, NS1-specific antibodies were found to protect against yellow fever encephalitis in mice[20]. Additionally, mAbs targeting the yellow fever virus NS1 protein protected monkeys against lethal challenge by invoking Fcγ-mediated effector functions[20–22]. Other work has shown that mAbs against West Nile virus NS1 protein prevent lethal infection in mice through Fcγ-receptor mediated

phagocytosis as well as an undetermined Fc-independent mechanism[23,24]. The dengue virus NS1 protein has been extensively studied in the context of antiviral immunity. Successful passive protection studies were performed in mice with NS1-specific monoclonal antibodies as well as protein and DNA plasmid-based vaccines[25–30]. Recently, dengue virus NS1 protein was shown to induce disruption of endothelial barriers in mice, which can also be prevented by vaccination with the NS1 protein[31]. Finally, an NS1-based vaccine for Zika virus was successfully tested in a mouse challenge model, proving that NS1-mediated immunity alone is sufficient for a protective vaccine[32]. These studies in many related flaviviruses suggest mAbs against the ZIKV NS1 protein are likely protective. In our work, we plan to determine if NS1-specific mAbs can provide protection in a murine model and whether this protection relies upon Fcγ-receptor effector functions.

Using a well-established protocol for the generation of fully human mAbs[33], we isolate ZIKV specific mAbs from the plasmablast compartment of a patient recently infected by ZIKV. The variable regions of the heavy and light chains of isolated plasmablasts are then sequenced, cloned, and recombinantly expressed. Here, we report the characteristics of four NS1-specific antibodies found in an individual with symptomatic ZIKV infection. Our data demonstrate that while NS1-specific mAbs do not neutralize virus in vitro, they can confer FcγR-mediated protection in vivo in a murine challenge model, which highlights the importance of NS1 epitopes in vaccine development.

## Results

**ZIKV NS1 specific antibodies are induced in humans.** To investigate the antibody response to ZIKV infection, we obtained plasma and isolated peripheral blood mononuclear cells (PBMCs) from a patient who was infected with ZIKV while traveling in Central America. The patient was likely not pre-exposed to dengue based on history and past travel. Blood was collected ten days after the patient tested positive for ZIKV RNA by RT-PCR. We adapted a published protocol to isolate Zika virus-specific mAbs from the plasmablast compartment of the infected patient[33,34].

Following single cell sorting of B cells, the variable regions of the immunoglobulins were sequenced, cloned into a human IgG1 expression vector, and subsequently expressed in HEK 293F cells as previously described[33,34]. The mAbs were then initially screened for reactivity to ZIKV-infected Vero cells by immunofluorescence. Vero cells were infected with one of two strains of Zika virus: either the African lineage MR766 which was isolated in Uganda from a rhesus macaque in 1947 and subsequently passaged in mice, or the Asian lineage PRVABC59, which was isolated in Puerto Rico from a human patient in 2015 and is representative of the current circulating strain. Additionally, we tested a DENV-3 isolate from the Philippines to determine if our antibodies cross-reacted with other flaviviruses. We found that three of the mAbs (AA12, EB9, and GB5) bound cells infected by both MR766 and PRVABC59 and one antibody (FC12) that bound cells infected by only PRVABC59 (Fig. 1a). None of our antibodies cross-reacted with DENV-3 infected cells. We next tested whether our antibodies bound by enzyme-linked immunosorbent assay (ELISA) to recombinant NS1 protein (Fig. 1b, c). We expressed and purified NS1 from both MR766 and PRVABC59 strains of ZIKV. As expected, three antibodies bound NS1 (AA12, EB9, and GB5) from both strains of ZIKV while FC12 only bound NS1 from the recent PRVABC59 ZIKV isolate. All mAbs were originally found to be of the IgG1 isotype and carried a low number of somatic mutations (Table 1). Neutralization activity was examined by microneutralization assays and

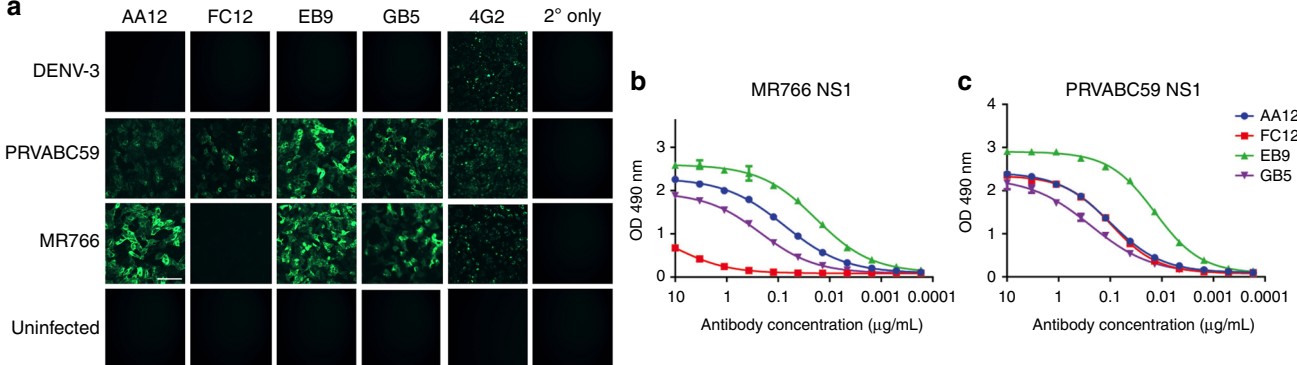

**Fig. 1** Human ZIKV specific-antibodies bind to both MR766 and PRVABC59 NS1 proteins. **a** Vero cells were infected with the indicated viruses at an MOI of 1 for 24 h. The cells were fixed with 0.5% paraformaldehyde and blocked with 5% non-fat milk. MAbs AA12, FC12, EB9, and GB5 were used at a concentration of 5 μg per mL and an anti-human antibody conjugated to Alexa Fluor 488 was used as a secondary antibody. The murine pan-flavivirus mAb 4G2 was used as a positive control and an anti-mouse antibody conjugated to Alexa Fluor 488 was used as a secondary antibody. Cells stained with mAb 4G2 were fixed and permeabilized using 80% acetone. **b, c** ELISA assays were performed using recombinant NS1 protein from either MR766 or PRVABC59 viruses to assess the binding activity of mAbs AA12, FC12, EB9, and GB5. ELISAs were performed in duplicates. Data plotted represent mean values and the standard error of the mean (SEM); a non-linear regression line was generated using GraphPad Prism 5. Scale = 100 μm

**Table 1 Antibody characteristics. VJ assignments, CDR3 sequences, % identity, and isotype for the antibody clones. IMGT/V-QUEST software was used to assign the germline reference for IGHV and IGLV and determine % identity to germline. ELISAs were done to determine reactivity and microneutralization assays were performed to determine neutralization activity at concentrations up to 100 μg per mL**

| Antibody | V-Gene | J-Gene | CDR3 | % Identity | Isotype | V-Gene | J-Gene | CDR3 | % Identity | Isotype | Reactivity to PRVABC59 NS1 | Reactivity to MR766 NS1 | Neutralization activity (IC₅₀) |
|---|---|---|---|---|---|---|---|---|---|---|---|---|---|
| AA12 | VH3-53 | JH3-02 | CARDRRGFDYW | 99% | IgG1 | VK1-39 | JK4-01 | CQQTYSTPLTF | 98% | Kappa | Yes | Yes | None detected |
| FC12 | VH3-53 | JH3-02 | CARGPVQLERRPLGAFDIW | 99% | IgG1 | VL3-1 | JL2-01 | CQAWDSSTVVF | 100% | Lambda | Yes | No | None detected |
| EB9 | VH3-53 | JH3-02 | CARWGGKRGGAFDIW | 100% | IgG1 | VK1-39 | JK2-01 | CQQSYSTPYTF | 98% | Kappa | Yes | Yes | None detected |
| GB5 | VH3-53 | JH3-01 | CARLIAAAGDYW | 99% | IgG1 | VK1-39 | JK1-01 | CQQSYSTPWTF | 98% | Kappa | Yes | Yes | None detected |

none of the NS1-specific mAbs exhibited neutralization activity against either PRVABC59 or MR766 (Table 1). The binding affinities of each antibody to PRVABC59 and MR766 were then determined using biolayer interferometry (Supplementary Table 1). We found that the binding constants using biolayer interferometry (Supplementary Table 1) were consistent with observed ELISA data (Fig. 1b, c). As expected, FC12 only bound NS1 from the recent PRVABC59 ZIKV isolate while AA12, GB5 and EB9 both bound potently to PRVABC59 or MR766.

**NS1 specific mAbs activate Fc mediated effector functions**. We next evaluated the ability of NS1-specific mAbs to engage in Fcγ-mediated effector functions. To model the activation of ADCC, we used a genetically modified Jurkat cell line expressing human FcγRIIIa and a luciferase reporter under a nuclear factor of activated T-cells (NFAT) promoter as a surrogate to examine the ability of these mAbs to engage and then activate Fcγ-mediated effector functions. First, we infected Vero cells with either MR766 or PRVABC59 ZIKV. Next, we added NS1-specific mAbs at concentrations ranging from 10 to 0.002 μg per mL. Consistent with our earlier ELISA results, three mAbs (AA12, EB9, and GB5) induced effector functions on both MR766 and PRVABC59 ZIKV infected cells and one antibody (FC12) induced effector functions only on PRVABC59 ZIKV (Fig. 2a, b).

Next, we determined whether transfection of NS1 is sufficient to activate Fc-FcγR effector functions by transfecting HEK

293T cells with an expression (pCAGGS) plasmid expressing NS1 from either MR766 or PRVABC59 ZIKV. After incubation with the same mAbs at the same concentrations, we found only two of our mAbs (AA12 and EB9) induced effector functions on both MR766 and PRVABC59 ZIKV transfected cells while the remaining two antibodies induced effector functions on cells transfected by NS1 from PRVABC59 ZIKV (Fig. 2c, d). The discrepancy observed in antibody GB5 may be due to a limited transfection efficiency of NS1 compared to NS1 being expressed from infected cells. Alternatively, the conformation of transfected NS1 from the PRVABC59 ZIKV isolate may limit the GB5 epitope as compared to the NS1 protein expressed by infected cells. Nevertheless, we can conclude that the surface NS1 protein in the absence of viral infection is sufficient to activate Fcγ-mediated effector functions induced by NS1-specific mAbs.

Lastly, we demonstrate that NS1-specific mAbs can direct the activation of human primary NK cells. Here, we infected Vero cells with PRVABC59 ZIKV at an MOI of 0.5. At 48 h post-infection, dilutions of NS1-specific IgG1 mAbs, an irrelevant mAb, or no mAb (starting at 20 μg per mL) in combination with isolated primary human NK cells were incubated with the ZIKV-infected Vero cells. After three hours, activation of NK cells was measured as percent of CD56+ cells expressing CD107a. As shown in Supplementary Figure 1, all NS1-specific mAbs can measurably activate CD107a expression of NK cells from two donors, ranging from 32 to 18% at 20 μg per mL. Of note, CD107a expression between the irrelevant IgG and baseline

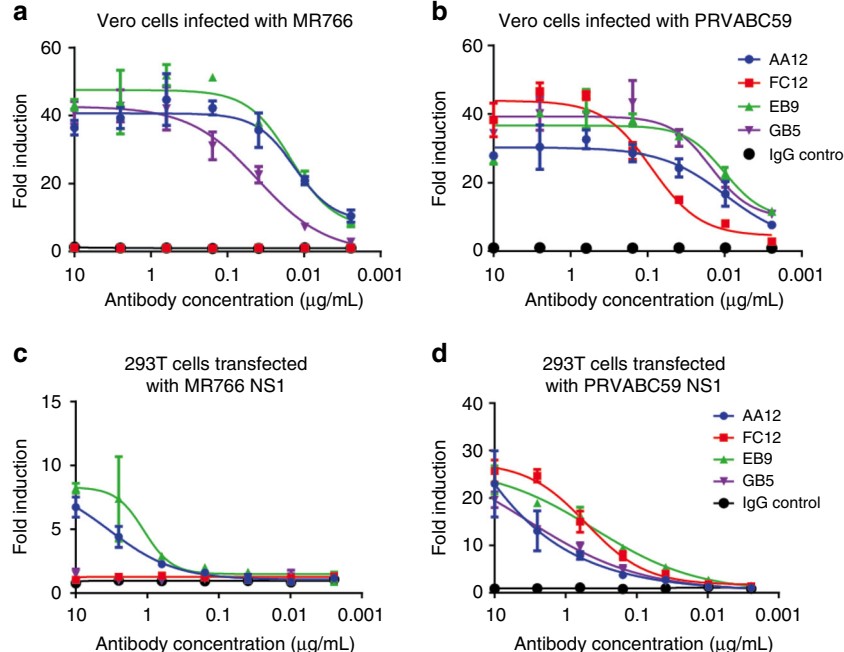

**Fig. 2** NS1-specific antibodies activate Fc-FcR effector functions in vitro. To examine the ability of NS1-specific antibodies to activate Fc-FcR mediated effector functions, **a, b** Vero cells were infected with MR766 and PRVABC59 Zika viruses or **c, d** HEK 293T cells were transfected with NS1 from MR766 and PRVABC59 Zika viruses. Infected Vero cells or transfected HEK 293T cells were used as targets for measuring antibody-mediated effector functions with a genetically modified Jurkat cell line expressing the human FcγRIIIa with an inducible luciferase reporter gene. Fold induction was measured in relative light units and calculated by subtracting background signal from wells without effector cells then dividing wells with antibody by wells with no antibody added. All mAbs were tested at a starting concentration of 10 μg per mL and were serially diluted four-fold. Assays were performed twice as technical duplicates and one of two replicates is shown. A non-linear regression best-fit curve was generated for each dataset using GraphPad Prism 5. Error bars represent SEM

control (with no mAb added) groups correlated with each other at around 14.06% for donor 1 and 22.6% for donor 2.

**NS1 specific mAbs do not enhance ZIKV or DENV infection.** We next assessed whether NS1-specific mAbs were able to enhance infection of target cells in vitro. ADE is commonly observed when an antibody that opsonizes but does not fully neutralize a virion facilitates infection of Fcγ-receptor bearing target cells. We measured ADE using a flow cytometry-based assay in which serial dilutions of monoclonal antibody or serum were mixed with PRVABC59 ZIKV and added to FcγR bearing K562 cells, which are typically non-permissible to ZIKV infection. After 48 h, cells were fixed and stained for the envelope protein using murine 4G2 antibody and the number of infected cells was determined by flow cytometry. We found that none of our NS1-specific mAbs enhance Zika infection in vitro (Fig. 3). In contrast, we observed a high level of ADE activity when K562 cells were infected in the presence of DENV-immune plasma, indicating the presence of cross-reactive antibodies between ZIKV and DENV consistent with published literature[14,35].

**mAb AA12 protects against lethal heterologous challenges.** To assess the ability of NS1-specific mAbs to protect against ZIKV disease in vivo, we performed lethal challenge experiments in a mouse model. As Zika virus does not replicate in wild-type mice, we used $Stat2^{-/-}$ mice which are permissive to Zika virus infection and can display clinical signs of disease[36]. We administered antibody AA12 intraperitoneally at 20 mg per kg two hours before challenge. Irrelevant mAb at 20 mg per kg was used as an isotype negative control. Mice were then infected with ten 50% mouse lethal doses ($10LD_{50}$) of the Zika virus strain MR766 and

weight loss, clinical scores, and survival were monitored daily. $Stat2^{-/-}$ mice were bred in-house and colony sizes were a limiting factor in the number of mice in each group to be tested. Therefore, we performed murine challenge studies as two or three independent replicates with at least three mice per treatment group and data were then pooled. Mice that received 20 mg per kg of AA12 showed minimal weight loss (Fig. 4a), significantly improved survival rate (Fig. 4b) and significantly lower clinical scores on 6 to 14 days post infection (dpi) (except 8 dpi) as compared to the IgG control (Fig. 4c). Specifically, mice that received 20 mg per kg of AA12 had an 83% survival rate while the IgG control all succumbed to disease (0% survival). To determine whether protection is relevant to the recent Zika virus outbreak or limited to the mouse-adapted MR766 strain, we tested our mAbs against a contemporary Panama 2015 strain (H/PAN/2015/CDC-259359) from the Asian Zika virus lineage (Fig. 4d, e). We challenged mice with the PAN/2015 isolate as this particular virus demonstrated consistent mortality at the dose of 500 PFU in $Stat2^{-/-}$ mice. In line with our previous data, AA12 was able to significantly protect animals from mortality at 10 mg per kg (63% survival), while the IgG control was not (0% survival). AA12 trended towards higher protection in the Panama 2015 strain challenge than the MR766 challenge likely due to the increased neurovirulence of the mouse-adapted MR766 strain. We also tested the two other monoclonal antibodies EB9 and FC12 in the same passive transfer challenge using PAN/2015 (Supplementary Figure 2). We show that EB9 protected 50% of the mice and FC12 only protected 25% of the mice compared to the IgG control group. As we show that AA12 provided high levels of protection against lethal challenge from both the historic African and more modern Asian lineage of Zika viruses as shown in the survival rates and disease clinical scores, this antibody was used for all

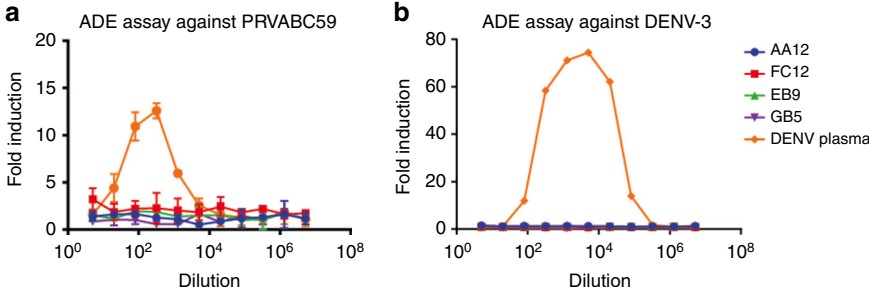

**Fig. 3** NS1-specific antibodies do not cause antibody-dependent enhancement (ADE) of infection in vitro. To examine whether enhancement of flavivirus infection in vitro is observed, ZIKV NS1-specific mAbs or pooled serum from a DENV positive donor were incubated with **a** PRVABC59 or **b** DENV-3 viruses and added to FcγR bearing K562 cells. All mAbs were tested at a starting concentration of 100 ng per mL and were serially diluted four-fold. DENV positive control sera was diluted five-fold initially and serially diluted four-fold. The assay was run in duplicate and fold induction was measured as number of infected cells as measured by flow cytometry divided by infected cells with no antibody or serum added. Sera were obtained through a screening of blood donations in Puerto Rico as described previously in Bardina et al.[14]. Plotted values represent mean value and standard deviation

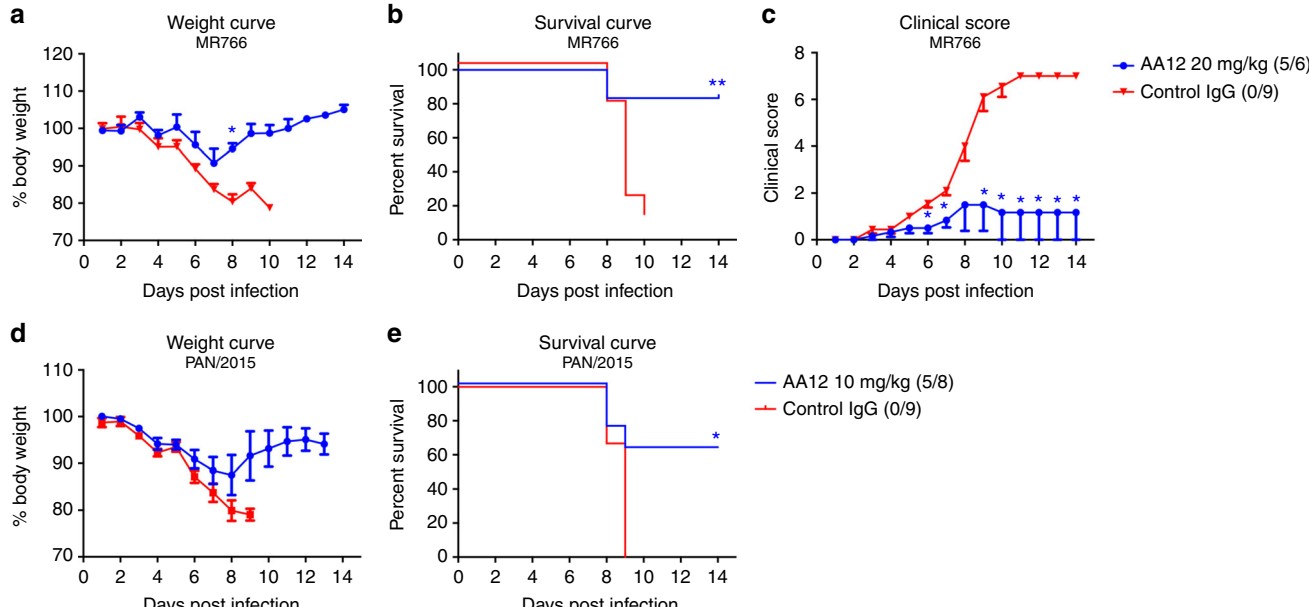

**Fig. 4** NS1-specific antibodies protect mice against lethal challenge in vivo. **a–c** Groups of 6–9 male and female B6.129-Stat2$^{-/-}$ mice were injected IP with 20 mg per kg of AA12 before a challenge with 10 mLD$_{50}$ of ZIKV MR766 intradermally. A mAb (CR9114) against influenza A virus was used as an IgG control. **d**, **e** Mice were treated with 10 mg per kg of AA12 or 10 mg per kg of isotype control before a challenge with 500 PFU of ZIKV PAN/2015 retro-orbitally. Weight loss was monitored daily. For mice infected with ZIKV MR766, clinical scoring was conducted using the pre-defined criteria with a maximum possible score of 7: impact on walking (1), unresponsiveness (1), left hind leg paralyzed (1), right hind leg paralyzed (1), left front leg paralyzed (1), and right front leg paralyzed (1). Deceased animals were awarded a score of 7. The ratios in the figures indicate the number of animals that survived challenge over total number of animals per group. Murine challenge studies were performed as two independent replicates with at least three mice per treatment group and data shown here were pooled. The Mantel-Cox and Gehan-Breslow-Wilcoxon tests were used to analyze statistical significance of survival between two groups. A multiple $t$-test and the Holm-Sidak method were used to determine statistical significance at each time point for the weight curve and the clinical score. Asterisk(s) indicates statistical significance of a group (*$p < 0.05$ and **$p < 0.005$) compared to control IgG. No significant differences between groups were detected in **d**

future experiments. Additionally, viral burden was measured in mice infected with PAN/2015 and treated with either 10 mg per kg of AA12 or IgG control (Supplementary Figure 3). On days 3 and 6 post-infection, mice were euthanized and spleens and brains were harvested and homogenized. Viral titers were then determined by plaque assay. Interestingly, no virus was detected in the brains of infected mice on day 3 post-challenge. However, on day 6, virus was detected in the brains of all three mice administered IgG control and only one of the three mice in the AA12 treatment group. On day 3, there was a significant reduction of viral load in the spleens of AA12 administered

mice compared to the control group, while only one mouse given AA12 had a detectable load of virus on day 6. Collectively, we demonstrate that administration of AA12 can significantly improve survival rates against two ZIKV strains, prevent disease severity and decrease viral titers in the spleens of mice.

**NS1 mediated protection is Fc dependent**. We next examined whether Fc-FcγR or Fc-complement interactions are required for providing protection in vivo. First, we cloned the heavy chain variable regions of AA12 into expression vectors containing the human IgG1 framework with the amino acid mutations L234A,

L235A, and P329G[37–39]. These mutations abolish the interaction of the Fc region with Fcγ receptors and complement proteins. To confirm that the mutations do not interfere with antigen binding, we tested both variants by ELISA (Fig. 5a, b). Both the WT and mutated form of AA12 bound to the MR766 and PRVABC59 NS1 proteins identically. These variants were tested again in an Fc-FcγR engagement assay, and, as expected, only the wild-type AA12 variant showed activity (Fig. 5c, d).

We then examined if our AA12 variant has any protective activity in vivo, by performing the same prophylactic passive transfer challenge as done previously with ZIKV MR766. We observed that while weight changes were not substantially different from the different groups (Fig. 6a), administration of wildtype AA12 (10 mg per kg) significantly improved the survival rate (~53%) and clinical score over mice receiving the AA12 with ablated Fc-FcγR and complement interactions (LALAPG) or the IgG control (Fig. 6b). Lastly, we demonstrate that Fc-FcγR or Fc-complement interactions are required to prevent onset of severe disease as wildtype AA12 can significantly decrease the clinical score (days 10–14) (Fig. 6c). Next, we tested the L234A, L235A mutations or the P329G mutation alone by introducing the LALA or PG point mutations into the AA12 antibody. Antibodies with either of these mutations alone had comparable binding affinity with the wildtype, were inactive in the Fc-FcγR engagement assay (Supplementary Figure 4) and were tested in a prophylactic passive transfer challenge with ZIKV MR766. The AA12 LALA was not protective and AA12 PG only protected 25% of mice. It is possible that the single point mutation in PG did not completely disrupt Fc-FcγR engagement and some Fc-FcγR effector functions remained resulting in modest protection in our challenge model. However, as shown in Fig. 6c, both AA12 LALA or AA12 PG failed to decrease onset of severe disease. Overall, our data strengthens our previous findings that Fc-FcγR effector functions remain critical in protection afforded by mAb AA12. We then wanted to determine whether complement was activated during infection by Zika virus. We tested whether treatment with the

AA12 variants ablated complement activation as compared to the wild-type AA12. We measured complement in serum at day 6 post-infection by ELISA in four of the mice used in the challenge from Fig. 6. We chose day 6 as mice typically begin to show clinical signs of infection and high viral burden at this point. All mice undergoing the challenge had higher levels of complement activation as compared to naïve uninfected STAT2[-/-] mice (Supplementary Figure 5). However, complement levels were elevated most in infected mice receiving the IgG control. It is therefore likely that these levels correlate most with morbidity and are activated by increased viral replication and the resulting heightened proinflammatory state. As wild-type AA12 protected mice against challenge more effectively than AA12 LALA or AA12 PG, it is unsurprising that complement levels were more elevated in the latter two groups.

## Discussion

Several other groups have isolated and characterized human monoclonal antibodies to ZIKV[8–12], however, the main focus of these studies was on potently neutralizing antibodies targeting E, the surface envelope glycoprotein present on the virion. While a strong neutralizing antibody response to structural viral proteins contributes to protection against infection and disease, less is known about non-neutralizing antibodies that target the non-structural proteins, such as NS1. Currently, there is a paucity of data on the protective efficacy of monoclonal antibodies that target the Zika virus NS1 protein. As many candidate flavivirus vaccines omit the NS1 component[40], it is possible that antibodies targeting the NS1 are overlooked in the protective immune response to ZIKV infection in humans. Of note, a recent study highlights the importance of incorporating NS1 in a multivalent vaccine against ZIKV[41]. Though Zika virus has one serotype with regards to a neutralizing response[42], less is known about whether non-neutralizing antibodies against NS1 are able to target multiple strains of the virus. NS1 is highly conserved amongst Zika virus strains, approaching 99.3% sequence identity[43]. The high

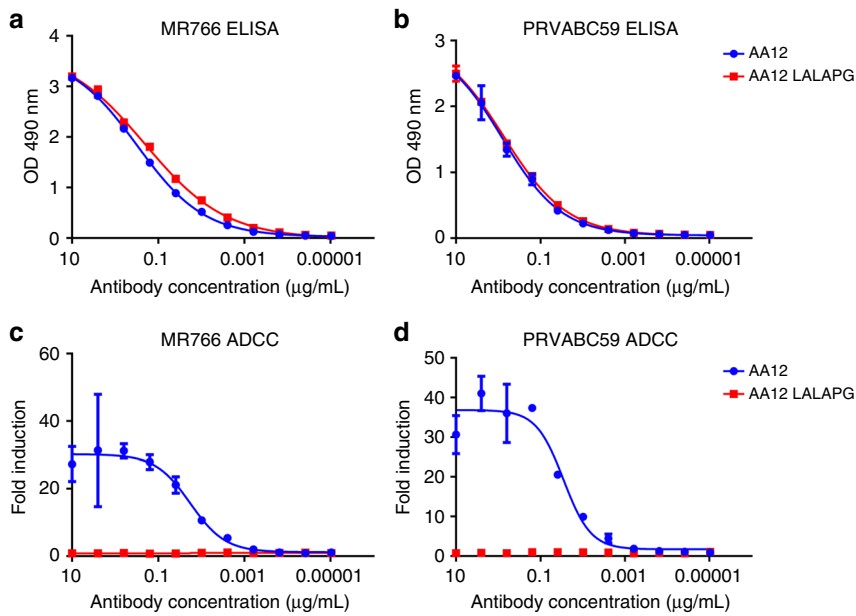

**Fig. 5** LALAPG mutation ablates Fc-FcR mediated effector functions without affecting affinity. The variable region of AA12 was cloned into human IgG1 or human IgG1 with L234A, L235A, and P329G mutations (LALAPG) in the backbone. **a, b** ELISA assays were performed using recombinant NS1 from either MR766 and PRVABC59 viruses to assess the binding activities of mAbs AA12 and AA12 LALAPG. **c, d** Fc-FcR mediated effector functions were tested on Vero cells infected with MR766 and PRVABC59 Zika viruses. AA12 was able to elicit Fc-FcR mediated effector functions while AA12 LALAPG was not. Assays were performed twice as technical duplicates and one of two replicates is shown. A non-linear regression best-fit curve was generated for each dataset using GraphPad Prism 5. Error bars represent SEM

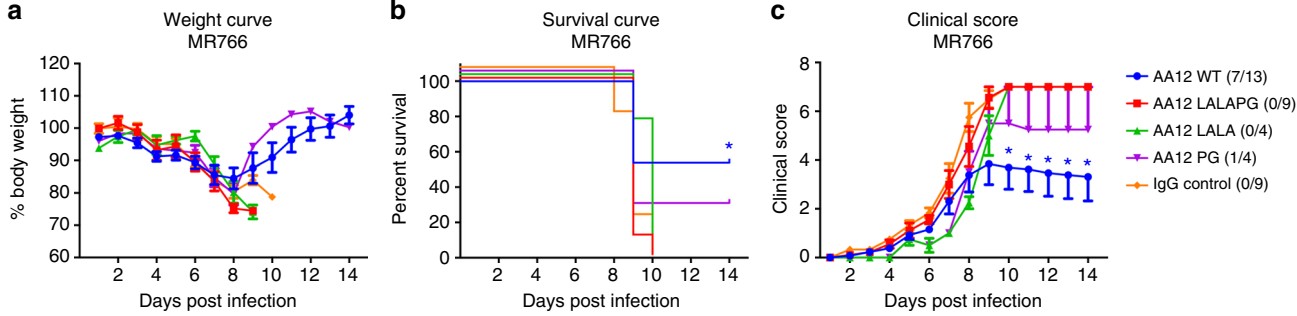

**Fig. 6** NS1-specific antibodies protect mice against lethal challenge in vivo in an Fc-dependent manner. **a–c** Groups of 4–13 male and female B6.129-Stat2$^{-/-}$ mice were injected IP with 10 mg per kg of wildtype AA12, AA12 LALAPG, AA12 LALA, or AA12 PG Fc-variants before a challenge with 10 mLD$_{50}$ of ZIKV MR766. The mAb CR9114 against influenza A virus was used as an IgG isotype control. Weight loss was monitored daily. Clinical scoring was conducted using the pre-defined criteria with a maximum possible score of 7: impact on walking (1), unresponsiveness (1), left hind leg paralyzed (1), right hind leg paralyzed (1), left front leg paralyzed (1), and right front leg paralyzed (1). Deceased animals were awarded a score of 7. The ratios in the figures indicate the number of animals that survived challenge over total number of animals per group. Murine challenge studies were performed as three independent replicates with at least four mice per treatment group and data shown here were pooled. Statistical analyses were performed using the Mantel-Cox and Gehan-Breslow-Wilcoxon tests for the survival curves and a multiple *t*-test and the Holm-Sidak method for the weight curve and the clinical score. Significance (*$p < 0.05$) is indicated compared to IgG control. No significant differences between the groups were detected in **a**

level of conservation in NS1 proteins implies an immune response targeting NS1 may protect against all circulating strains and is a good candidate target for a vaccine. In fact, a recent study demonstrated that a ZIKV NS1-based vaccine using a Modified Vaccinia Ankara (MVA) vector is protective against a heterologous ZIKV challenge in mice[32]. Notably, the vaccine was given to wild-type mice who were subsequently challenged intracerebrally. As passive transfer studies have not been conducted, it is unclear which arm of the adaptive immune response, cell-mediated or humoral antibody immunity, contributed most to protection against disease. Our studies build on previous work and indicate that NS1-specific mAbs contribute to protection and should play in important role in the formulation of novel flavivirus vaccines.

In the present study, we isolated plasmablasts from the PBMCs of a ZIKV-infected individual around 15–20 days after infection. Cloning of the variable regions of the antibody sequences isolated from plasmablasts revealed four NS1-specific mAbs that can bind to the recent Puerto Rico (PRVABC59) isolate of ZIKV. Interestingly, only the antibody FC12 is unable to bind the historic Uganda (MR766) strain of ZIKV, which has been isolated from rhesus macaques in 1947 and subsequently passaged in mice. While the ZIKV NS1 protein is highly conserved across many strains, the finding that one (FC12) of four mAbs isolated recognized only a recent ZIKV strain suggest that there may be different immunodominant regions of NS1 that vary between isolates. By isolating and characterizing more NS1-specific mAbs, the antigenic regions of the NS1 protein can be mapped and used for incorporating the NS1 protein in candidate ZIKV vaccines. With biolayer interferometry, we show the affinity of our antibodies to be between 10$^{-7}$ and 10$^{-8}$ molar, which suggests a moderate level of affinity. However, the bivalent manner by which NS1-specific mAbs bind to homodimeric NS1 suggest that avidity may play a role in increasing their biological function in vivo. Though the calculated affinity is lower than many potently neutralizing antibodies, it must be noted that these antibodies have lower levels of somatic hypermutation. We speculate that we may find more NS1-specific antibodies with higher affinities and levels of somatic hypermutation in the memory B cell compartment of the same individual or in individuals who have had repeated exposures to the virus.

Vaccines that elicit NS1-specific mAbs do not risk inducing antibody-dependent enhancement of disease (ADE). In contrast, the recent Dengvaxia® vaccine, which induced antibodies to the structural components of dengue virus, caused an increased risk of severe disease in flavivirus naïve children, resulting in the suspension of the sale and distribution of the vaccine in the Philippines[44,45]. To date, there are no clear cases involving ADE induced by ZIKV infection in humans. However, passive transfer of DENV or WNV immune plasma to immunocompromised mice has resulted in more severe disease progression upon ZIKV infection[14]. Additionally, ZIKV-induced monoclonal antibodies can enhance infection of DENV in vitro[8]. This is a major concern as ZIKV and dengue viruses are closely related, share the same mosquito vector, and impact the same geographic regions. Our studies suggest that as the NS1-specific mAbs are unable to enhance viral uptake in vitro, an NS1-based vaccine will be a safer alternative to current flavivirus vaccine preparations.

Using a murine challenge model, we demonstrate that NS1-specific mAbs can prevent death and disease in vivo. Our studies use B6.129-Stat2$^{-/-}$ mice that are challenged intradermally with MR766 or retro-orbitally with PAN/2015. Though the two ZIKV sequences are highly conserved, the viruses were isolated 68 years apart and display different phenotypes in mice. Therefore, both strains were tested in our studies. Infection with the MR766 strain represents a stringent challenge, inducing a higher level of inflammatory cytokines and severe neurological symptoms when mice are infected intradermally[36]. Mice infected with the Asian lineage strain PAN/2015 were infected retro-orbitally to display consistent lethality in our challenge models. We found that mAb AA12 is able to significantly improve the survival rates of mice at doses of 20 mg per kg (83%) or 10 mg per kg (53%) during MR766 challenge and at a dose of 10 mg per kg (63%) during PAN/2015 challenge. Moreover, both doses of wildtype AA12 significantly decreased disease as measured by clinical score. Treatment of AA12 was also found to greatly reduce viral burden in the spleens of mice infected with PAN/2015 at day 3 post-infection. Moreover, we show that EB9 and FC12 trend towards partial protection against the recent PAN/2015 isolate of ZIKV at 10 mg per kg. Future experiments will determine the optimal range by which full protection is achieved.

Fcγ-mediated protection induced by non-neutralizing or poorly neutralizing antibodies has been found to play an important role in the context of many other viral infections including influenza A virus[46–49], and it is unsurprising that these functions may protect against Zika virus disease. To explore

whether Fcγ-mediated immunity is required for protection against ZIKV challenge, we cloned the heavy chain variable region of AA12 into an expression plasmid with the mutations L234A, L235A, and P329G in the Fc region[37–39]. These mutations resulted in ablated Fcγ-effector functions as measured by a surrogate reporter assay. The finding that the mutant AA12 mAb (AA12-LALAPG) is unable to protect mice against lethal challenge suggest that activation of Fcγ-mediated effector functions is the mechanism by which protection is achieved. Additionally, AA12 antibodies with the L234A, L235A, or P329G mutations alone are also unable to protect mice against lethal challenge. To determine whether complement plays a role in reduction of viral burden, we measured C3 levels in mice undergoing lethal challenge. We found that C3 levels are elevated in mice treated with AA12 LALAPG, AA12 PG, and IgG Control compared to mice treated with wild-type AA12. As wild-type AA12 suppresses viral burden and decreases disease severity, similar suppression of complement activation is seen. However, as mutant antibodies are unable to protect against lethal challenge, complement levels are increased—likely correlating with increased levels of disease and viral burden.

Notably, a high concentration of mAbs is required to confer protection against lethal challenge. As these mAbs are non-neutralizing and target a nonstructural protein, sterilizing immunity is not achieved. Though NS1-specific antibodies may not protect against initial infection, these antibodies limit disease severity as measured by decreasing weight loss and clinical score in antibody-treated animals. The induction of Fcγ-mediated protection by NS1-specific antibodies may therefore be an overlooked correlate of protection in the hunt for promising Zika virus vaccines. A complete vaccine that elicits not only neutralizing but also NS1-specific antibodies may increase protection against Zika virus disease in humans.

A lack of established diagnostics also hampers ZIKV virus vaccine development. Often, neutralizing antibody titers are used as a readout if viral RNA levels are not detected. Testing serum by plaque reduction neutralization tests may be additionally complicated by high levels of dengue virus cross-reactive antibodies. As the NS1 protein is highly conserved amongst ZIKV strains but only exhibits 55% identity with dengue virus[43], testing for NS1-specific antibodies may lead to better ZIKV diagnostics. Recent studies demonstrate a rapid NS1-based antigen test using monoclonal antibodies[50–52]. We add to this work by reporting a highly specific antibody (FC12) only able to recognize the more recent ZIKV isolate. We suspect that this antibody may provide a high-level of sensitivity and specificity in detecting serum levels of ZIKV NS1 in patients infected by recent outbreaks.

It is also notable that all four NS1-specific antibodies isolated from this patient had the same VH3⁻53/JH3 rearrangement but with different light chains (Table 1). This same rearrangement was found in NS1-specific antibodies isolated from two different patients from a separate recent study[53]. Therefore, this rearrangement is found in many expanded B cell clones across the human population that target Zika virus NS1. Further investigation into how this germline rearrangement affects antibody binding to NS1 is warranted. This is also the first report of recurring antibodies that share the same IGV genes in the context of Zika virus NS1.

Teratogenic effects of ZIKV on the developing fetus in pregnant mothers is a major concern in the ongoing epidemic. In fact, it was the causal relationship between ZIKV infection during pregnancy and microcephaly that led the WHO to declare Zika virus a "public health emergency of international concern". In light of our work, future studies should examine the prevention of viremia and maternal-fetal transfer of virus by NS1-specific mAbs or NS1-based vaccines.

In summary, our work helps to further dissect the components of the antibody response against Zika virus. We have highlighted the importance of mAbs targeting the NS1 protein which can dramatically protect against disease and death in a murine challenge model. Furthermore, we demonstrate that NS1 antibody-based protection against ZIKV disease is Fc-mediated. Lastly, the lack of ADE induction as measured by an in vitro assay suggests an NS1-based vaccine can reduce the risk of severe disease in flavivirus naïve patients as compared to a structural protein-based vaccine.

## Methods

**Cells and viruses**. Human embryonic kidney (HEK) 293T cells (American Type Culture Collection; ATCC Cat. No. CRL-1573) and African green monkey kidney cells (Vero; ATCC; ATCC CCL-81) were grown in Dulbecco modified Eagle medium (DMEM, Gibco) supplemented with 10% fetal bovine serum (FBS) (Hyclone) and antibiotics (100 units per mL penicillin–100 μg per mL streptomycin [Pen-Strep]; Gibco). Human embryonic kidney Expi293F cells (Gibco) were grown in Expi293 expression media. MR766 virus (Rhesus/1947/Uganda BEI NR-50065), PRVABC59 virus (2015/Puerto Rico; BEI NR-50684) and PAN/2015 virus (H/PAN/2015/CDC-259359) were obtained from BEI resources. Fcγ-receptor expressing K562 cells were obtained through ATCC (Cat. No. CCL-243). Pan-flavivirus antibody 4G2 was obtained through ATCC D1-4G2-4-15 (ATCC® HB-112™). ZIKV were propagated in Vero cells in 1× Minimum Essential Medium (MEM); after 72 h post-infection (hpi), cell culture supernatants were harvested, aliquoted and stored at −80 °C until use.

**Human plasmablast isolation**. Plasmablasts were isolated at approximately two weeks after onset of symptoms. Plasmablasts (CD19+/CD3-/CD20−/CD38high/CD27high) were isolated and monoclonal antibodies were generated[33] in accordance with the Icahn School of Medicine at Mount Sinai Institutional Review Board. Briefly, Ficoll density (GE Healthcare) centrifugation was performed to isolate the buffy coat, and peripheral blood mononuclear cells (PBMCs) were single-cell sorted onto freshly made catch buffer (50 μL of 1 M Tris pH 8 and 125 μL of Rnasin in 5 mL of RNAse-free water) on 96-well plates using a BD FACSARIA III. Reverse transcription reactions were performed to generate cDNA[33,34]. Two nested PCRs incorporating IgG-specific, IgA-specific, IgM-specific, kappa-specific, and lambda-specific primers were performed on the cDNA to amplify heavy and light chains. The International Immunogenetics Information System software, (http://www.imgt.org/IMGT_vquest/vquest) was used to view productive immunoglobulin sequence rearrangements. Sixteen Zika virus antibodies were isolated from one patient, four of which are NS1-specific and further characterized in this study. All four NS1-specific antibodies, AA12, EB9, FC12 and GB5 have the VH3-53/JH3 heavy chain and are of the IgG1 isotype (Table 1).

**Recombinant human antibodies**. The human heavy (VH) and light (VL) variable regions of the antibodies AA12, EB9, GB5, and FC12 were amplified by PCR and cloned into human IgG1 and kappa or lambda mammalian expression vectors, respectively (pFUESss-CHIg-hIgG1, pFUESss-CLIg-hK or pFUSEss-CLIg-hl2; Invivogen). The L234A, L235A, and P329G (LALAPG) mutations in the IgG1 heavy chain were introduced by site-directed mutagenesis. The variable region of the heavy chain of AA12 was then cloned into the modified expression vector to make AA12-LALAPG. Wild-type or LALAPG antibodies (mutated IgG1 heavy chain with wild-type kappa chain) were expressed and purified[33].

**Recombinant ZIKV NS1**. The NS1 gene segments from MR766 virus (Rhesus/1947/Uganda Accession: NC_012432) and PRVABC59 virus (2015/Puerto Rico Accession: KU501215) were human codon optimized using Integrated DNA Technologies Codon Optimization Tool (http://www.idtdna.com/CodonOpt) and modified to contain a C-terminal hexahistidine-tag. NS1 gene segments were subcloned into the expression plasmid pCAGGS using restriction endonucleases NotI and XhoI (New England Biosciences) and inserted into the digested plasmid by homologous recombination (In-Fusion, Takara) to construct pCAGGS-MR766-NS1 and pCAGGS-PRVABC59-NS1. To generate recombinant NS1 proteins, 30 mL of Expi293 cells were transfected with 30 μg of pCAGGS-MR766-NS1 or pCAGGS-PRVABC59-NS1 plasmids and 81 μL of expifectamine reagent as per manufacturer's instructions. After 120 h, supernatants were cleared by low-speed centrifugation and incubated with Ni-NTA resin overnight at 4 °C. The resin-supernatant mixture was then passed over 10 mL polypropylene columns (Qiagen). The retained resin was washed four times with 15 mL of washing buffer (50 mM $Na_2HCO_3$, 300 mM NaCl, 20 mM imidazole, pH 8) and protein was eluted with elution buffer (50 mM $Na_2HCO_3$, 300 mM NaCl, 300 mM imidazole, pH 8). The eluate was concentrated using Amicon Ultracell (Millipore) centrifugation units with a cut-off of 10 kDa and buffer was changed to phosphate buffered saline (PBS) of pH 7.4. Protein concentration was quantified using Pierce Bicinchoninic Acid Protein Assay Kit (Thermo Scientific) with a bovine serum albumin standard curve. Purified soluble NS1 proteins were resolved in a reducing and denatured

SDS-PAGE gel (in monomeric forms of around 45 kDa and homodimeric forms of around 90 kDa) and visualized using SimplyBlue SafeStain (Thermofisher, Inc.).

**Enzyme-linked immunosorbent assay**. Plates were coated with recombinant ZIKV NS1 at 2 µg per mL in pH 9.41 carbonate buffer overnight at 4 °C. After blocking in 5% non-fat (NF) milk for 1 h, mAbs were incubated at a starting concentration of 10 µg per mL and serially diluted 3-fold and incubated 2 h at room temperature. Horseradish peroxidase (HRP)-conjugated goat anti-human IgG antibody (AP504P; Millipore Sigma) was used to detect binding of the mAbs, followed by development with HRP substrate (Sigmafast OPD; Sigma-Aldrich). Reactions were stopped by addition of 3 M HCl and absorbance was measured at 490 nm on a microplate spectrophotometer (BioRad). Experiments were performed in duplicates and repeated twice. Graphpad Prism 5 was used to visualized the mean values and the standard error of the mean (SEM) and generate a non-linear regression curve.

**Immunofluorescence**. Vero cells were infected with ZIKV MR766, ZIKV PRVABC59 or dengue virus type 3, Philippines/H87/1956 with a multiplicity of infection (MOI) of 1. After 24 h post-infection, the monolayer of Vero cells was fixed with 0.5% of paraformaldehyde (PFA)/1× PBS. Cells were blocked with 5% nonfat milk for 30 min at room temperature. Blocking buffer was then discarded and NS1-specific mAbs were added at a concentration of 5 µg/mL in nonfat milk. Primary antibodies were incubated for 2 h at room temperature after which the monolayer was washed three times with 1× PBS. An anti-human or anti-mouse IgG secondary antibody conjugated to Alexa Fluor 488 (ThermoFisher) diluted (1:000) in nonfat milk was added to the monolayer and incubated in the dark at RT for 1 h. The monolayer was then washed three times with 1× PBS. Cells were then visualized using an inverted fluorescent microscope (Olympus IX70).

**Microneutralization assay**. To assess the in vitro neutralizing activity of the mAbs we performed a microneutralization (MN) assay. Three-fold serially diluted antibody (starting at 100 µg per mL) in serum-free minimum essential medium (MEM) was mixed with an equal volume of virus (100 $TCID_{50}$) and incubated for 1 h at room temperature. Monolayers of Vero cells were washed once with PBS and the virus/antibody mixture was added to the cells and incubated for 1 h at 37 °C. After the infection, the virus/antibody mixture was removed and replaced with serum-free MEM with antibody added at the appropriate dilution. The cells were then incubated at 37 °C for 72 h. Cytopathic effect (CPE) was scored at three days post infection and $IC_{50}$ was quantified by the Reed and Muench method.

**Antibody dependent effector functions**. For experiments involving infected cells, Vero cells were seeded on 96-well flat white-bottom plates (Corning) and infected after 24 h with ZIKV (MR766 or PRVABC59) at an MOI of 0.01. For experiments involving transfected cells, HEK 293T cells were seeded onto poly-D-lysine coated 96-well flat white-bottom plates (Corning). After 24 h, the cells were transfected with 100 ng per well of expression plasmid encoding NS1 from MR766 or PRVABC59. At 16 h post-transfection or 40 h post-infection, the medium was removed and 25 µL of assay buffer (RPMI 1640 with 4% low-IgG FBS) was added to each well. Then mAbs were added in a volume of 25 µL at 30 µg per mL and serially diluted fourfold in assay buffer (in duplicate). The mAbs were then incubated with the transfected or infected cells for 30 min at 37 °C. Genetically modified Jurkat cells expressing the human FcγRIIIa with a luciferase reporter gene under the transcriptional control of nuclear factor-activated T cells (NFAT) promoter were added at $7.5 \times 10^4$ cells at 25 µL per well, which is approximately a 1:2 ratio of target cells to effector cells, followed by incubation for another 6 h at 37 °C (Promega). Bio-Glo Luciferase assay reagent was added after 6 h and luminescence was quantified using a plate reader. Fold induction was measured in relative light units and calculated by subtracting background signal from wells without effector cells then dividing wells with antibody by with no antibody added. Specifically, fold induction was calculated as follows: $(RLU_{induced} - RLU_{background})/(RLU_{uninduced} - RLU_{background})$. The mean values and SEM were reported and a nonlinear regression curve was generated using GraphPad Prism 5.

Alternatively, we measured antibody-dependent effector functions by detecting activation of primary human natural killer (NK) cells (expression of CD107a) as previously described[47]. Briefly, Vero cells were seeded at $2 \times 10^4$ cells per well in 96-well cell culture-treated plates and infected with PRVABC59 ZIKV with an MOI of 0.5 (adjusted for cell growth overnight). At 48 hpi, growth media from infected Vero cells were aspirated and incubated with dilutions (50 µL total volume) of NS1-specific mAbs (diluted in 1× Iscove's media supplemented with 10% FBS) starting at 20 µg per mL, serially diluted 3-fold and incubated at 37 °C, $CO_2$ for 1.5 h. A human mAb specific for the influenza B virus hemagglutinin, II2C7, was used as an irrelevant mAb and a group containing no mAb was used as background control. Human NK cells were isolated (Lymphoprep; Stemcell Technologies, Inc.) from buffy coat donors (San Diego Blood Center) through negative selection (EasySep Human NK cell isolation kit; Stemcell Technologies, Inc.) and $8 \times 10^5$ CD56+ cells/well (in 50 µL 1× Iscoves's media supplemented with 10% FBS; effector cells to target ratio of 2) were subsequently added to the ZIKV-infected Vero cells and mAb mixture (total volume of 100 µL). Cells were incubated for 3 h at 37 °C, $CO_2$. Cells were then washed with wash buffer (1X PBS/1% BSA)

and stained with CD56-FITC (Clone B159 BD Biosciences; 5 µL per 1e6 cells) and CD107a-PE (Clone H4A3 BD Biosciences; 20 µL per 1e6 cells) for 15 min at 4 °C (in the dark). Samples were then resolved in a BD FACS ARIA II flow cytometer sorter (BD Biosciences) and analyzed using FlowJo 10.5.0. Experiments were performed in duplicates and the means/standard error were graphed using GraphPad Prism 5.

**$K_D$ determination**. Biolayer interferometry assays were performed with an Octet RED instrument (ForteBio, Inc.) to determine $K_D$ values. Purified recombinant NS1 was loaded onto a Ni-NTA biosensor (ForteBio, Inc.) in kinetics buffer (1× PBS pH 7.4, 0.01% BSA, 0.002% Tween-20) for 3 min. To determine $k_{on}$, association was measured for 3 min by exposing the sensors to seven concentrations of antibody diluted in kinetics buffer. To determine $k_{off}$, dissociation was measured for 3 min in kinetics buffer. $K_D$ values were calculated as the ratios of $k_{off}$ to $k_{on}$. We used a 2:1 binding model to reflect two identical binding sites of homodimeric NS1 proteins. K1 and K2 reflect the kinetics constants of the first and seconding binding interaction between the mAb and a homodimeric NS1.

**Antibody dependent enhancement of infection**. Enhancement of ZIKV or DENV infection was measured using a flow-cytometry-based assay[14]. Serial dilutions of purified monoclonal antibody were mixed with ZIKV (PRVABC59 MOI of 1) for 1 h at 37 °C in RPMI 1640 media supplemented with 10% FBS, 2 mM L-glutamine, and antibiotics (100 units per mL penicillin–100 µg per mL streptomycin [Pen-Strep]; Gibco). The mixture was then added to K562 cells in 96-well U bottom plates. After two days, cells were fixed with 4% PFA/1X PBS, permeabilized with PBS containing 0.2% BSA and 0.05% saponin and stained with 4G2 pan-flavivirus anti-envelope antibody (1 µg per mL) for 1 h at RT. Cells were then incubated with goat anti-mouse IgG conjugated to phycoerythrin (1 µg per mL; Invitrogen) for 1 h at RT. The number of infected cells was determined by flow cytometry using a FACS Caliber and analyzed using FlowJo2 software version 10.1.r7. Area under the curve was calculated using GraphPad Prism.

**Passive transfer studies**. All animal experiments were performed in an animal biosafety level 2 plus facility in accordance with the Icahn School of Medicine at Mount Sinai and the University of California, Riverside Institutional Animal Care and Use Committees (IACUC). Groups of 5–9 male and female B6.129-$Stat2^{-/-}$ mice (kindly provided by Dr. Christian Schindler) were passively transferred with 20 or 10 mg per kg AA12, or 10 mg per kg AA12-LALA, AA12-PG or AA12-LALAPG antibody intraperitoneally. Control mice received the human anti-influenza antibody CR9114[54] at a dose of 10 mg per kg. Mice were challenged intradermally with 10 $LD_{50}$ ZIKV MR766 or retro-orbitally with 500 PFU of ZIKV PAN/2015 and evaluated for 14 days. Mice were monitored daily for weight and clinical signs. Clinical scoring was conducted using the pre-defined criteria with a maximum possible score of 7: impact on walking, unresponsiveness, left hind leg paralyzed, right hind leg paralyzed, left front leg paralyzed, and right front leg paralyzed. Deceased animals were given a score of 7[14,55]. Animals that showed more than 25% weight loss or full paralysis were humanely euthanized. Experiments were conducted with a balanced amount of male and female mice and with an even distribution of mice from different litters whenever possible. To determine statistical significance, the Mantel-Cox and Gehan-Breslow-Wilcoxon tests were used for survival curves and a multiple t-test and the Holm-Sidak method utilized to analyze the weight curve and clinical scores. Asterisk(s) on graphs indicates statistical significance (*p value < 0.05 and **p value < 0.005) of a group compared to the IgG control group.

**Viral titers**. Tissue samples were harvested from infected mice, placed in PBS, and homogenized using ceramic beads. ZIKV quantification was conducted via plaque assay. Briefly, Vero cells were plated in 24-well plates and infected after 24 h with dilutions of virus made in serum-free 1× MEM medium. Infectious medium was aspirated and 600 µL of methylcellulose agar equivalent medium was added to each well. At day 4 post-infection plates were fixed with 4% PFA for 1 h at RT, washed, and stained with 4G2 in milk at 5 µg per mL. Secondary anti-mouse HRP was then added at 1:5000 in milk and the assay was resolved with TrueBlue Peroxidase Substrate (VWR). Plaques were manually counted and plaque forming units (PFU) per mL of homogenized tissue was calculated.

**Complement ELISA**. A mouse complement C3 ELISA kit (Abcam: ab157711) was used as per the manufacturer's instructions to measure C3 levels in the serum of infected or naïve mice. Briefly, serum was diluted 1:50,000 and pipetted into designated wells. In parallel, a standard curve was generated from known concentrations of C3. The plate was incubated for 20 min, washed, and a 1× enzyme-antibody conjugate was added. After incubation for twenty minutes and additional washing, TMB substrate was added and absorbance was measured at 450 nm on a microplate spectrophotometer (BioRad).

**Study approval**. An IRB approved written informed consent was obtained from the patient prior to study participation. No further demographic data are included here in order to protect the participant's privacy.

**Statistical analysis**. Results from multiple experiments are presented as mean ± SEM. Student's *t*-tests were used to test for statistical differences between mean values. Data were analyzed with GraphPad Prism 7 software and *p*-values of <0.05 were considered statistically significant.

## Data availability

The sequences of the immunoglobulin variable regions for AA12 (VH accession no. MH931170; VL accession no. MH931171), EB9 (VH accession no. MH931172; VL accession no. MH931173), GB5 (VH accession no. MH931174; VL accession no. MH931175), and FC12 (VH accession no. MH931176; VL accession no. MH931177) have been deposited in GenBank. The authors declare that all other data supporting the findings of this study are available within the article and its Supplementary Information files, or are available from the authors upon request.

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

## Acknowledgements

We thank B. Fulton and P. Leon for valuable assistance in plasmablast isolation. We also acknowledge the Flow Cytometry Shared Resource Facility. This work was supported by 1R21AI129477 (F.K.), R01AI120998 (V.S), R21AI133649 (M.J.E.). M.J.E. holds an Investigators in Pathogenesis of Infectious Disease Award from the Burroughs Wellcome Fund. G.S.T. is supported by a J. Craig Venter Institute start-up fund and U19A110819. The animal studies performed in this study are in accordance with the Institutional Animal Care and Use Committee of the Icahn School of Medicine at Mount Sinai and the University of California, Riverside.

## Author contributions

M.J.B. and G.S.T. conceived the study. M.J.B., J.D., H.D. F.B., J.A.B., F.O.A., V.H.L.G., and G.S.T., designed and performed the experiments. M.C.B.G. isolated the blood sample. M.J.B. and G.S.T. isolated and characterized the human mAbs. M.J.E., V.S., J.K.L., F.K., R.H., P.P., alongside the other authors analyzed the data. M.J.B. and G.S.T. wrote the manuscript and all other authors reviewed and approved the final version.

## Additional information

**Competing interests:** The authors declare no competing interests.

