## [Peer Review File · Nature Communications]

Reviewers' Comments:

Reviewer #1:

Remarks to the Author:

The manuscript entitled "Human antibodies targeting the Zika virus NS1 protein provide protection against disease in a mouse model" is very well written and presents results showing the importance of antibodies against the NS1 protein of Zika virus in protection. However there are some points that could be better explained or explored.

Experiments to determine the binding affinity of the different antibodies should be better explained, either in Materials and Methods or Results sections, as well as in Table 2. The purified recombinant NS1 protein used in this assay was from the African or Asian lineage? What is the difference between K1 and K2? The legend of this table should also inform what X2 and R2 are. Besides, authors should explore more the results and discuss the correlation between data obtained with the binding affinity and ELISA assays.

The materials and methods of the "Microneutralization assay", pg 12, should be reviewed. After 1 hour of infection, cells were maintained for 72 hours in serum-free medium? And incubated with more antibody (line 15)?

The materials and methods of "Antibody-dependent effector function" assay could also be more explicative. What is the ratio of effector : target cell, for instance?

In the experiments of protection, authors should comment why they chose the AA12 monoclonal antibody and not the EB9, for instance, to perform all the analysis. Also in this section, they should inform why they used the antibody concentration of 10mg/Kg in all the subsequent analyses instead of 20mg/Kg that led to high survival rates.

The statistical analyses are missing in all graphs, especially in experiments of protection. All the experiments of protection were performed only once? What are the confiability of results? In Figure 4E, inoculation of 10mg/Kg of the AA12 monoclonal antibody led to 60% survival, while in Figure 6B, only 44% of mice inoculated with the same antibody concentration survived challenge.

Minor points:

Introduction, pg 3, line 6, please correct "study of four patients" and not "patent".

In the description of results, section "Antibodies targeting Zika...", line11, authors should include the information of in which country the Asian lineage PRVABC59 was isolated.

In section "Monoclonal antibody AA12 provides protection...", in line 6, include 10 LD50, instead of mLD50.

In the first section of Materials and Methods, "Cells and Viruses", authors should inform what are the Expi293F cells. Human embryonic kidney cells?

In Materials and Methods as well in legends of Figures 4 and 6, authors informed that the maximum possible score for clinical signs of infection was 6. However, when we look at the graphs the maximum score is 7 (including death). Authors should correct it in the text.

Reviewer #2:

Remarks to the Author:

The study by Beiley at al. entitled "Human antibodies targeting the Zika virus NS1 protein provide protection against disease 1 in a mouse model" describes the generation and in vitro and in vivo characterization of four NS1 specific human monoclonal antibodies from plasmablasts isolated from a Zika infected patient. The study is well written, interesting and timely. It is also of broad interest both from a basic scientific perspective as well as from a vaccine development perspective against this significant human pathogen.

However, in my opinion there are several major concerns and additional details that should be addressed prior to publication.

Major:

1. A major concern is that no information is provided about experimental rigor and reproducibility throughout the manuscript. As far as can be gleaned from reading the manuscript, each experiment was only performed once. This is especially concerning when it comes to the in vivo analyses, which are inherently highly variable and require independent repeats for verification of results.
2. For all of the in vivo experiment it is a major shortcoming that no viral titers are provided, either kinetically or at least at the peak of infection.
3. A more extensive discussion of the papers having shown NS1-mediated protection against other flaviviruses are warranted. Currently they are referenced, but only very briefly mentioned.
4. Figure 4. It is unclear why the clinical score is only shown for the MR766 and not for the Pan/2015 challenge. It is unclear why the figure uses the contemporary Pan/2015 strain while all other experiments uses the PRVABC59 strain. It is unclear why the Pan/2015 experiment only uses the lower dose of mAb, when significantly better results were obtained at the higher dose with the MR766 challenge. It is unclear why the MR766 challenge dose is given as 10xLD50, while the Pan/2105 challenge dose is only provided as a PFU per inoculum. The figure legend does not mention challenge route. This information is given in the methods, and in fact the MR766 challenge is done intradermally, while the Pan/2105 is done retro-orbitally. This needs to be made more clear in the legend or the figure itself. The most significant shortcoming of this figure however, is the fact that it seems the experiment was only performed once.

Other:

1. No details in terms of the clinical history of the patient that the antibodies were derived from is provided. Was the patient likely pre-exposed to dengue?
2. No information is provided as to the plasmablast response of the patient. What was the Ig isotype distribution of the isolated plasmablasts? I.e of the ZIKV specific cells, how many cells were IgG, IgA or IgM secreting?
3. Was the single cell PCR performed to identify IgG/IgM/IgA or only IgG? While the resulting antibodies were all expressed in an IgG1 vector, the original isotype of the cells are not discussed. Given the lack of dengue cross reactivity, the analyzed donor is likely a primary infection, meaning that the IgM response might be of great interest. Finally, the IgG VH gene used for all these antibodies are the same VH3-53/JH3 rearrangement, but with different light chains. While there is only four antibodies in all, this should be mentioned and discussed.

4. How many total antibodies were isolated from this patient to obtain the four NS1 antibodies? i.e. what was the frequency of NS1 specific mAbs?
5. Figure 1: Is the axis on the ELISA assay supposed to be ng, or microgram? A/B/C in the legend does not match the figure.
7. Figure 2. Figure legend states that the error bars shows SEM, but no info on the number of experimental replicates are provided, nor how many independent repeats were performed.
8. Figure 3. Legend does not spell out the a and b panels. The sera used is stated to be from a previous publication, but it is not referenced.
9. Figure 5. The legend is mislabeled between the ELISA assays and the functional assays. No information regarding reproducibility is provided.
10. Figure 6. This seems to be a single experiment as well. Why is this experiment done at a suboptimal mAb concentration? It is unclear why these experiments use the old MR766 isolate instead of a more recent contemporary isolate.

Reviewer #3:

Remarks to the Author:

In this study, Bailey and colleagues characterized a set of human monoclonal antibodies (mAbs) specific to the Zika virus (ZIKV) NS1 protein. These antibodies, isolated from a patient infected with ZIKV, were non-neutralizing and capable of activating Fc-FcγR-mediated effector functions in vitro and did not enhance ZIKV or dengue virus infection in vitro. In addition, one of the mAbs, AA12, provided protection against lethal challenge with ZIKV isolates MR766 and PAN/2015 in a mouse model, which was dependent on FcγR engagement of host cells. This is a well-conceived and straightforward study, where the conclusions are generally supported by the data presented. However, certain additions and clarifications are necessary.

Major Comments

1. Authors used a LALAPG mutated version of mAb AA112 to establish that Fc receptor functions are critical for protection by anti-NS1 antibodies. However, by using this mutant, one cannot determine whether i) Fc-FcγR interaction, ii) Fc-complement interaction or iii) both interactions are required for protection by mAb AA12. Single mutations (i.e., AA12-LALA and AA12-PG), in addition to AA12-LALAPG, are needed to fully address the question that the authors themselves raised regarding whether Fc-FcγR or Fc-complement interactions are required for providing protection in vivo.
2. The experiments using genetically modified Jurkat cells expressing human FcγRIIIa to assess Fc-mediated effector functions are informative; however, these experiments do not fully demonstrate that these responses can be triggered directly by ZIKV NS1. Therefore, authors should provide additional evidence showing that ADCC can be elicited by anti-NS1 mAbs in an antigen-specific manner. To further define the protective mechanisms of the tested antibodies, authors should determine whether these mAbs are capable of activating ADCC and possibly complement in vitro. Even though FcγRIIIa (CD16) is well known to mediate ADCC, the use of Jurkat FcγRIIIa-expressing cells only demonstrate that an Fc effector-mediated function was activated but not that ADCC occurs with actual antigen recognition by the antibodies and cell killing with effector cells. Additional ADCC assays need to be performed to demonstrate whether ZIKV-infected cells or NS1-transfected cells are in fact

being killed in the presence of anti-ZIKV NS1 mAbs. For instance, this could be done by using co-cultures of effector cells such as NK cells (CD16+) in combination with ZIKV-infected cells/NS1-transfected cells as target cells to measure the levels of either cytotoxicity of the target cells (infected/transfected cells) or NK cell degranulation.

3. Related to the previous issue, authors also need to establish if in vivo complement activation is required for protection by anti-NS1 mAbs. This could be tested by using complement-deficient mice (crossed to B6.129-Stat2^{-/-} mice) for in vivo protection experiments and/or determining the levels of complement proinflammatory molecules (e.g., C3a and C5a) that may circulate in infected mice. In the latter scenario, use of mutant mAbs (see comment 1) should reverse the activation of complement components.

4. In Figures 5 and 6, the authors show a wild-type AA12 mAb triggered Fc-FcR-mediated effector functions in vitro and protected mice against lethal challenge in vivo compared to an AA12 LALAPG mutation. However, was this protective mechanism related to greater activation of complement? Could this process lead to lower viremia levels in infected mice? Authors should clarify this. The addition of this data is necessary to understand the mechanism of protection in infected mice.

5. Figure 2a and b need to be explained in more detail in the Figure legend and in the Results text. What does the fold induction mean with ZIKV-infected Vero cells -- versus what control? The Methods section is lacking a clear description of this experiment as well.

Minor Comments

1. Authors should clarify why they used different strains of the Asian Zika virus lineage for in vitro assays (PRVABC59) and for in vivo protection experiments (H/PAN/2015/CDC-259359).

2. Why was only mAb AA12 used for in vivo experiments? This is puzzling since the in vitro data suggest that other mAbs, such as EB9, are more efficient at binding NS1 and activating Fc-FcR effector functions.

3. Table 2 with the affinity for these antibodies seems like data that could be made supplementary. There is almost no discussion of the Table 2 results in the text.

4. Can the authors comment on whether more than four mAbs were generated from this one patient; were these four mAbs chosen from a larger group?

Reviewers' comments:

Reviewer #1 (Remarks to the Author):

The manuscript entitled “Human antibodies targeting the Zika virus NS1 protein provide protection against disease in a mouse model” is very well written and presents results showing the importance of antibodies against the NS1 protein of Zika virus in protection. However there are some points that could be better explained or explored.

Experiments to determine the binding affinity of the different antibodies should be better explained, either in Materials and Methods or Results sections, as well as in Table 2. The purified recombinant NS1 protein used in this assay was from the African or Asian lineage?

We have amended the figure legend and table to clarify that we tested the NS1 protein from both the African and Asian - lineages. Please see the amended figure legend for Table 2 which we have now moved to the supplemental data section in Table S1.

What is the difference between K1 and K2?

We added an explanation of K1 and K2 and how they relate to the 2:1 binding motif used to fit the data. Please see lines 460 to 462 and the figure legend of Table S1.

The legend of this table should also inform what X2 and R2 are.

We added a brief description of X² and R² and acceptable values for this experiment in the table legend. Please see an amended Table S1 and its accompanying figure legend.

Besides, authors should explore more the results and discuss the correlation between data obtained with the binding affinity and ELISA assays.

We have made the affinity information supplementary (Table S1) and also discuss the correlations between data observed by Octet with ELISA data in the text. Please see lines 130 to 133.

The materials and methods of the “Microneutralization assay”, pg 12, should be reviewed. After 1 hour of infection, cells were maintained for 72 hours in serum-free medium? And incubated with more antibody (line 15)?

This is correct, antibody was added after the infection to detect for neutralization at a post-infection step and as a more rigorous test to determine whether NS1-specific antibodies were able to neutralize virus. The methods section was re-written for clarity. Please see lines 413 to 418.

The materials and methods of “Antibody-dependent effector function” assay could also be more explicative. What is the ratio of effector : target cell, for instance?

The ratio of target cells to effector cells was added in the materials and methods section and the section was re-written for clarity. Please see lines 425 to 433.

In the experiments of protection, authors should comment why they chose the AA12 monoclonal antibody and not the EB9, for instance, to perform all the analysis. Also in this section, they should inform why they used the antibody concentration of 10mg/Kg in all the subsequent analyses instead of 20mg/Kg that led to high survival rates.

We have included additional challenge data demonstrating EB9 and FC12 also confer partial protection in mice. AA12 was chosen for further analysis as it exhibited clear of protection in preliminary studies and high concentrations of antibody produced in transfected cells allowed for high dosages (20mg/kg) tested in mice. Please see new Figure S2 and lines 186 to 196 and 292 to 293. As the point mutations introduced may have ablated some but not all effector functions we wanted to use a minimally protective dose for AA12 to ensure Fc-FcR effector functions were needed for protection.

The statistical analyses are missing in all graphs, especially in experiments of protection. All the experiments of protection were performed only once? What are the confiability of results? In Figure 4E, inoculation of 10mg/Kg of the AA12 monoclonal antibody led to 60% survival, while in Figure 6B, only 44% of mice inoculated with the same antibody concentration survived challenge. The experiments shown in the figures are pooled data from a combination of several experiments all performed more than once. This is clarified in all the figure legends. The discrepancy in survival is likely due to the difference in lethality of a mouse-adapted MR766 virus used in figure 6 compared to contemporary PAN/2015 isolate in the mouse model (discussed in lines 191 to 193 and 292 to 293). Alternatively, this may also reflect the inherent efficacy of specific NS1 antibodies in protecting against different strains of ZIKV. As shown in an additional experiment,

we performed in Figure S2, mAb AA12, FC12 and EB9 (at 10 mg/kg) demonstrate variable protection against PAN/2015. We have addressed this in the manuscript text. Please see lines 186 to 196 and new Figure S2.

Minor points:

Introduction, pg 3, line 6, please correct “study of four patients” and not “patent”.

Corrected. Please see line 72.

In the description of results, section “Antibodies targeting Zika...”, line 11, authors should include the information of in which country the Asian lineage PRVABC59 was isolated.

Corrected. Please see line 119.

In section “Monoclonal antibody AA12 provides protection...”, in line 6, include 10 LD50, instead of mLD50.

Corrected. Please see line 179.

In the first section of Materials and Methods, “Cells and Viruses”, authors should inform what are the Expi293F cells. Human embryonic kidney cells?

Yes, they are human embryonic kidney cells. This detail was added to the methods section. Please see line 344.

In Materials and Methods as well in legends of Figures 4 and 6, authors informed that the maximum possible score for clinical signs of infection was 6. However, when we look at the graphs the maximum score is 7 (including death). Authors should correct it in the text.

Thank you for the correction. The maximum clinical score is indeed 7. Please see line 482; and the legends in Figures 4 and 6.

Reviewer #2 (Remarks to the Author):

The study by Beiley et al. entitled “Human antibodies targeting the Zika virus NS1 protein provide protection against disease 1 in a mouse model” describes the generation and in vitro and in vivo characterization of four NS1 specific human monoclonal antibodies from plasmablasts isolated from a Zika infected patient. The study is well written, interesting and timely. It is also of broad interest both from a basic scientific perspective as well as from a vaccine development perspective against this significant human pathogen.

However, in my opinion there are several major concerns and additional details that should be addressed prior to publication.

Major:

1. A major concern is that no information is provided about experimental rigor and reproducibility throughout the manuscript. As far as can be gleaned from reading the manuscript, each experiment was only performed once. This is especially concerning when it comes to the in vivo analyses, which are inherently highly variable and require independent repeats for verification of results.

Murine challenge data are pooled from two independent replicates with at least three mice per treatment group and this is made clear in the figure legends and the text. Please see lines 180 to 183 and the figure legends for Figure 4 and 6.

2. For all of the in vivo experiment it is a major shortcoming that no viral titers are provided, either kinetically or at least at the peak of infection.

Thank you for your recommendation. Viral titers were performed on mice infected with PAN/15 in two different tissue samples (brain and spleen) at two different time points (3 and 6 days post infection). We have included the data in the text (lines 197 to 204) and in the new supplemental Figure S3.

3. A more extensive discussion of the papers having shown NS1-mediated protection against other flaviviruses are warranted. Currently they are referenced, but only very briefly mentioned.

Thank you for your recommendation. In the introduction, we have expanded upon the studies with other flaviviruses that determine whether NS1-specific antibodies are protective. Please see lines 80 to 97.

4. Figure 4. It is unclear why the clinical score is only shown for the MR766 and not for the Pan/2015 challenge. It is unclear why the figure uses the contemporary Pan/2015 strain while all other experiments use the PRVABC59 strain. It is unclear why the Pan/2015 experiment only uses the lower dose of mAb, when significantly better results were obtained at the higher dose with the MR766 challenge. It is unclear why the MR766 challenge dose is given as 10xLD50, while the Pan/2105 challenge dose is only provided as a PFU per inoculum. The figure legend does not mention challenge route. This information is given in the methods, and in fact the MR766 challenge is done intradermally, while the Pan/2105 is done retro-orbitally. This needs to be made more clear in the legend or the figure itself. The most significant shortcoming of this figure however, is the fact that it seems the experiment was only performed once.

Thank you for your insightful comments. We have provided additional clarification in the text. Clinical scores for the PAN/2015 challenges for a repeat challenge have been updated and included in the supplemental challenge data (Figure S2) and in the text (please see lines 193 to 196). The doses of AA12 tested were protective in mice undergoing challenge with PAN/2015. We additionally discuss in line 296 that higher protection will likely be achieved with a higher dose. PAN/2015 was given as plaque forming units per inoculum because the reported dose consistently resulted in 100% lethality in control mice (please see lines 188 to 189). Challenge routes were clarified in the figure legends and in the methods section. The murine challenge studies were performed as two independent replicates with at least three mice per treatment group and additional challenge data is provided in supplementary figure S2. Please see lines 186 to 196.

Other:

1. No details in terms of the clinical history of the patient that the antibodies were derived from is provided. Was the patient likely pre-exposed to dengue?

This was added to the results section. Please see line 111 and 123 (Figure 1a).

2. No information is provided as to the plasmablast response of the patient. What was the Ig isotype distribution of the isolated plasmablasts? I.e. of the ZIKV specific cells, how many cells were IgG, IgA or IgM secreting?

As indicated in our methods (line 352 to 363), we single-sorted cells gated off the CD27+/CD38+ compartment. While IgG-, IgA- and IgM-specific primers were included during the PCR amplification steps (please see lines 358 to 360), all NS1-specific antibodies were found to be of the IgG1 isotype (please see Table 1 and lines 127 and 363).

3. Was the single cell PCR performed to identify IgG/IgM/IgA or only IgG? While the resulting antibodies were all expressed in an IgG1 vector, the original isotype of the cells are not discussed.

We have indicated that all NS1-specific antibodies are IgG1 (please see Table 1 and lines 127 and 363).

Given the lack of dengue cross reactivity, the analyzed donor is likely a primary infection, meaning that the IgM response might be of great interest.

A total of 16 ZIKV-specific antibodies were isolated from the plasmablast compartment. However, we found all antibodies to be of the IgG isotype. None of the NS1-specific antibodies are of the IgM isotype so the IgM response was not able to be fully analyzed in this study.

Finally, the IgG VH gene used for all these antibodies are the same VH3-53/JH3 rearrangement, but with different light chains. While there is only four antibodies in all, this should be mentioned and discussed.

Thank you for the correction. We have indicated the detail in lines 362 to 363 and further discussed these findings in lines 321 to 326 of the discussion section.

4. How many total antibodies were isolated from this patient to obtain the four NS1 antibodies? i.e. what was the frequency of NS1 specific mAbs?

A total of 16 ZIKV-specific antibodies were isolated from the plasmablast compartment. Four (reported in this manuscript) out of the 16 are NS1-specific. This was added to the text in lines 361 to 363.

5. Figure 1: Is the axis on the ELISA assay supposed to be ng, or microgram? A/B/C in the legend does not match the figure.

We have corrected the figure axis and legend to display micrograms. Please see a revised Figure 1 and its figure legend.

7. Figure 2. Figure legend states that the error bars shows SEM, but no info on the number of experimental replicates are provided, nor how many independent repeats were performed.

We have corrected the figure legend to explain the number of experimental replicates and independent repeats performed. Please see corrected legend for Figure 2.

8. Figure 3. Legend does not spell out the a and b panels. The sera used is stated to be from a previous publication, but it is not referenced.

We have corrected the figure legend to spell out the different viruses used and added the reference to the serum used for the study. Please see amended legend for Figure 3.

9. Figure 5. The legend is mislabeled between the ELISA assays and the functional assays. No information regarding reproducibility is provided.

We have corrected the figure axis and legend to display micrograms consistently throughout the panels. We have also corrected the figure legend to explain the number of experimental replicates and independent repeats performed.

10. Figure 6. This seems to be a single experiment as well. Why is this experiment done at a suboptimal mAb concentration? It is unclear why these experiments use the old MR766 isolate instead of a more recent contemporary isolate.

Murine challenge data are pooled from TWO independent replicates and this is made clear in the figure legends and the text. Furthermore, as the point mutations introduced may have ablated some but not all effector functions we wanted to use a minimally protective dose for AA12 to ensure Fc-FcR effector functions were needed for protection and did the head-to-head comparison of the ablated AA12 with the wild-type AA12 at the same concentration. MR766 represents a relevant and stringent test of efficacy of antibodies as mice are able to be infected with an intradermal infection (see lines 286 to 291).

Reviewer #3 (Remarks to the Author):

In this study, Bailey and colleagues characterized a set of human monoclonal antibodies (mAbs) specific to the Zika virus (ZIKV) NS1 protein. These antibodies, isolated from a patient infected with ZIKV, were non-neutralizing and capable of activating Fc-Fc γ R-mediated effector functions *in vitro* and did not enhance ZIKV or dengue virus infection *in vitro*. In addition, one of the mAbs, AA12, provided protection against lethal challenge with ZIKV isolates MR766 and PAN/2015 in a mouse model, which was dependent on Fc γ R engagement of host cells. This is a well-conceived and straightforward study, where the conclusions are generally supported by the data presented. However, certain additions and clarifications are necessary.

Major Comments

1. Authors used a LALAPG mutated version of mAb AA12 to establish that Fc receptor functions are critical for protection by anti-NS1 antibodies. However, by using this mutant, one cannot determine whether i) Fc-Fc γ R interaction, ii) Fc-complement interaction or iii) both interactions are required for protection by mAb AA12. Single mutations (i.e., AA12-LALA and AA12-PG), in addition to AA12-LALAPG, are needed to fully address the question that the authors themselves raised regarding whether Fc-Fc γ R or Fc-complement interactions are required for providing protection *in vivo*. Antibodies with single AA12-LALA and AA12-PG mutations were developed and found that antibodies with either variants were unable to elicit Fc-FcR effector functions in our reporter assay. We proceeded with a murine challenge study and show that while 75% of mice survive infection when treated with AA12 wild-type, no mice survive when treated with the AA12-LALA and only 25% of mice survive in the AA12-PG treatment group, which is not significantly different than the IgG control. This underscores our previous findings that Fc-FcR effector functions are essential for protection *in vivo* as either mutation, which disrupts these functions, is unable to protect mice from lethal challenge. Please see the new figure S4 and lines 219 to 227.

2. The experiments using genetically modified Jurkat cells expressing human Fc γ RIIIa to assess Fc γ -mediated effector functions are informative; however, these experiments do not fully demonstrate that these responses can be triggered

directly by ZIKV NS1. Therefore, authors should provide additional evidence showing that ADCC can be elicited by anti-NS1 mAbs in an antigen-specific manner.

In Figure 2c and 2d we have addressed whether NS1 protein alone can activate Fc-FcγR effector functions. We transfected HEK 293T cells with an expression (pCAGGS) plasmid expressing NS1 from either MR766 or PRVABC59 ZIKV. Here, we use the same assay as done in Figure 2a and 2b using genetically modified Jurkat cells to measure Fc-FcγR engagement. Based on our data we can conclude that the surface NS1 protein in the absence of viral infection is sufficient to activate Fcγ-mediated effector functions induced by NS1-specific mAbs. Therefore, ADCC can be elicited by anti-NS1 mAbs in an antigen-specific manner.

To further define the protective mechanisms of the tested antibodies, authors should determine whether these mAbs are capable of activating ADCC and possibly complement in vitro. Even though FcR3A (CD16) is well known to mediate ADCC, the use of Jurkat FcγR3A-expressing cells only demonstrate that an Fc effector-mediated function was activated but not that ADCC occurs with actual antigen recognition by the antibodies and cell killing with effector cells. Additional ADCC assays need to be performed to demonstrate whether ZIKV-infected cells or NS1-transfected cells are in fact being killed in the presence of anti-ZIKV NS1 mAbs. For instance, this could be done by using co-cultures of effector cells such as NK cells (CD16+) in combination with ZIKV-infected cells/NS1-transfected cells as target cells to measure the levels of either cytotoxicity of the target cells (infected/transfected cells) or NK cell degranulation.

We performed additional experiments demonstrating the ability of primary human NK cells to upregulate CD107a upon recognition of target cells in an antibody-dependent manner. We now demonstrate that these NS1-specific mAbs engage effector cells with two approaches: 1) using genetically modified Jurkat cells expressing human FcγR3A (please see lines 136 to 153; Figure 2c–d) and 2) primary human NK cells freshly isolated from PBMCs (please see the new figure S1, lines 154 to 161, and lines 437 to 453).

3. Related to the previous issue, authors also need to establish if in vivo complement activation is required for protection by anti-NS1 mAbs. This could be tested by using complement-deficient mice (crossed to B6.129-Stat2^{-/-} mice) for in vivo protection experiments and/or determining the levels of complement proinflammatory molecules (e.g., C3a and C5a) that may circulate in infected mice. In the latter scenario, use of mutant mAbs (see comment 1) should reverse the activation of complement components.

We did not have access to complement-deficient mice to cross with our *Stat2*^{-/-} mice. However, we were able to determine the levels of complement proinflammatory molecules in infected mice. Using the same challenge study shown in the new supplemental figure S4, we tested serum from mice at day 6 post infection. We chose this day because mice typically begin to show acute signs of ZIKV infection as measured by clinical score between days 5-7 and our viral titer data. We can therefore infer that this at this point virus is being actively cleared. A C3 ELISA was performed which quantitatively measures complement C3 in mouse serum. We find that all mice undergoing the ZIKV challenge had elevated complement levels compared to naïve uninfected *Stat2*^{-/-} mice. Additionally, there is a trend towards higher complement levels in groups of mice who do not survive the challenge versus the AA12 WT treatment group. From this we can conclude that the point mutations in AA12 likely do not affect the levels of complement produced by mice, as complement is produced even when control IgG is administered and is actually lowest when AA12 WT is administered. Based on our data, it is likely that complement is activated due to viral replication and the resulting proinflammatory state. As AA12 is able to clear virus via Fc-FcR effector functions, the treatment group had the lowest level of complement activation. Please see the new Figure S5 and lines 227 to 236.

4. In Figures 5 and 6, the authors show a wild-type AA12 mAb triggered Fc-FcR-mediated effector functions in vitro and protected mice against lethal challenge in vivo compared to an AA12 LALAPG mutation. However, was this protective mechanism related to greater activation of complement? Could this process lead to lower viremia levels in infected mice? Authors should clarify this. The addition of this data is necessary to understand the mechanism of protection in infected mice.

The new supplemental Figure S5 addresses whether levels of complement are different in mice receiving different treatment. Based on our data there is little evidence to suggest that AA12 WT results in a higher level of complement that results in the more rapid clearance of viral infection as complement levels were higher in all mice infected with ZIKV as compared to naïve uninfected *Stat2*^{-/-} mice. Finally, we believe our study addresses the role of Fc-FcR in AA12 providing protection against ZIKV infection in mice. Please see the new Figure S5 and lines 227 to 236.

5. Figure 2a and b need to be explained in more detail in the Figure legend and in the Results text. What does the fold

induction mean with ZIKV-infected Vero cells -- versus what control? The Methods section is lacking a clear description of this experiment as well.

Fold induction was measured as relative light units and calculated by subtracting the background signal from wells without effector cells then dividing wells with antibody by wells with no antibody added. This is clarified in the figure legends as well as the methods section. Please see lines 433 to 434.

Minor Comments

1. Authors should clarify why they used different strains of the Asian Zika virus lineage for in vitro assays (PRVABC59) and for in vivo protection experiments (H/PAN/2015/CDC-259359).

The different strains represent strains available to us at the time that provided reliable *in vivo* murine data. When doing preliminary studies, PRVABC59 did not consistently demonstrate lethality in mice and a new, relevant strain PAN/2015 was used because lethality was consistent in the in vivo experiments were performed. We briefly discuss this in the text. Please see lines 187 to 188.

2. Why was only mAb AA12 used for in vivo experiments? This is puzzling since the in vitro data suggest that other mAbs, such as EB9, are more efficient at binding NS1 and activating Fc-FcR effector functions.

We have included additional challenge data demonstrating EB9 also confers protection in mice. Despite more potent in vitro binding and Fc-FcR effector functions, mice receiving AA12 or EB9 demonstrated comparable *in vivo* efficacy. AA12 was used predominately because the high concentrations of antibody produced in transfected cells allowed for high dosages (20mg/kg) tested in mice. Additional challenge data with antibodies AA12, EB9, and FC12 is provided in new supplementary figure S2. Please see lines 193 to 196.

3. Table 2 with the affinity for these antibodies seems like data that could be made supplementary. There is almost no discussion of the Table 2 results in the text.

We have made the affinity information supplementary (Table S1) and also discuss the correlations between data observed by Octet with ELISA data in the text (Please see lines 130 to 133).

4. Can the authors comment on whether more than four mAbs were generated from this one patient; were these four mAbs chosen from a larger group?

A total of 16 antibodies were isolated from the plasmablast compartment that was discovered to be ZIKV-specific, with 4 (reported in this manuscript) out of the 16 being NS1-specific. This was added to the text in lines 361 to 363.

Reviewers' Comments:

Reviewer #1:

Remarks to the Author:

Manuscript accepted.

Authors have performed new experiments in order to answer questions posed by reviewers. Some initial experiments could have been performed differently, but apparently there are limitations, as for example the availability of viral samples and an efficient experimental murine model for such tests. Nevertheless, results are robust and are worth being published.

Reviewer #2:

Remarks to the Author:

I am satisfied that the authors have addressed all my previous concerns.

Prior to publication the full length sequence of the antibodies included herein should be submitted to GenBank, or a rationale provided for why this data is not included in the manuscript.

Reviewer #3:

Remarks to the Author:

In the revised version of the manuscript (plus response to reviewers document), most of my comments were addressed; in some cases, the experiments were not exactly those requested but the authors did attempt to experimentally address the issues. Thus, overall, the responses are adequate. One request, however, is to add to the Discussion section some mention of the discussion in Results on lines 238-243. In other words, some version of the concept that viral replication activates complement, C3 levels correlate with disease, and the AA12 antibody seems to suppress C3 levels and complement activation, the authors postulate, by suppressing viral replication. The mutant Mabs do not, so it seems that some other FcR function is critical. I don't need to see the manuscript again assuming this is addressed.

Reviewer Comments

Reviewer #1 (Remarks to the Author):

Manuscript accepted.

Authors have performed new experiments in order to answer questions posed by reviewers. Some initial experiments could have been performed differently, but apparently there are limitations, as for example the availability of viral samples and an efficient experimental murine model for such tests. Nevertheless, results are robust and are worth being published.

We thank the reviewer for their helpful comments.

Reviewer #2 (Remarks to the Author):

I am satisfied that the authors have addressed all my previous concerns.

Prior to publication the full length sequence of the antibodies included herein should be submitted to GenBank, or a rationale provided for why this data is not included in the manuscript.

We thank the reviewer for their helpful comments. The GenBank accession numbers for the sequences of the immunoglobulin variable regions will be added under the Data Availability section (lines 545 to 547) prior to final publication of this work (highlighted in yellow).

Reviewer #3 (Remarks to the Author):

In the revised version of the manuscript (plus response to reviewers document), most of my comments were addressed; in some cases, the experiments were not exactly those requested but the authors did attempt to experimentally address the issues. Thus, overall, the responses are adequate. One request, however, is to add to the Discussion section some mention of the discussion in Results on lines 238-243. In other words, some version of the concept that viral replication activates complement, C3 levels correlate with disease, and the AA12 antibody seems to suppress C3 levels and complement activation, the authors postulate, by suppressing viral replication. The mutant Mabs do not, so it seems that some other FcR function is critical. I don't need to see the manuscript again assuming this is addressed.

We thank the reviewer for their helpful comments. We have added to the Discussion section that include interpretation of our results of the complement assay (lines 314 to 319; highlighted in yellow).